# Predicting the Effect of Surface Waviness on Fatigue Life of a Wire + Arc Additive Manufactured Ti-6Al-4V Alloy

**DOI:** 10.3390/ma16155355

**Published:** 2023-07-30

**Authors:** Muhammad Shamir, Xiang Zhang, Abdul Khadar Syed, Wayne Sadler

**Affiliations:** Centre for Manufacturing and Materials, Coventry University, Coventry CV1 5FB, UK; m.shamir@amrc.co.uk (M.S.); abdul.syed@coventry.ac.uk (A.K.S.); ac1991@coventry.ac.uk (W.S.)

**Keywords:** WAAM, surface waviness, surface roughness, fatigue, bending test, durability, fracture mechanics

## Abstract

This paper reports the effect of as-deposited surface conditions on the fatigue strength of an additively manufactured titanium alloy, Ti-6Al-4V (WAAM Ti64). First, the local stress concentration caused by the surface waviness was quantified using a metrology technique and computer modelling. Fatigue tests were conducted under bending loads at a cyclic load ratio of 0.1. The applicability of two predictive methods was the focus of this study. The traditional notch stress method was unable to predict the correct S–N curve trend slope, which could be attributed to the early crack initiation from the troughs on the as-built surface, with crack propagation being the dominant failure mechanism. By treating the troughs as small cracks, the fracture mechanics approach delivered good predictions at every applied stress level. Surface machining and polishing may not always be practical or required; it depends on the applications and service load levels. This research demonstrated that the fracture mechanics approach can be used for predicting the fatigue life of WAAM titanium alloys in as-built conditions and, hence, can be a tool for decision making on the level of surface machining.

## 1. Introduction

Wire and arc additive manufacturing (WAAM) is a directed energy deposition (DED) AM process that is capable of producing near-net shape and large-scale metallic structures at low manufacturing costs, high deposition rates, and virtually no porosity defects in high-strength titanium parts [1,2,3]. Among the materials built via WAAM and other AM processes, titanium alloy Ti6Al4V (Ti64) is the most studied owing to its high production cost via conventional manufacturing routes and its wide applications in the aerospace, biomedical, and energy sectors [1,2,3]. The yield and ultimate tensile strengths of WAAM Ti64 are comparable with conventional wrought materials, which are considerably higher than the cast materials, and meet the requirements for tensile strength as set by the ASTM F2924 standard for AM Ti64 [4]. However, the elongation value in WAAM Ti64 is about 40% lower than that of wrought materials due to the presence of columnar grains, which is caused by repeated thermal cycles and fine *α* laths due to faster cooling rates in the process [4]. The fatigue strength of WAMM Ti64 is similar to wrought materials at 10^7^ cycles and is higher than the cast materials [4]. Despite the advantages in material fabrication and comparable mechanical performance with conventional materials, one of the challenges is the poor surface finish in the as-deposited condition. The characteristic surface undulations, which are described as surface waviness within this paper, can cause a considerable reduction in the fatigue strength due to the stress concentration arising from the troughs.

Load-bearing parts produced by AM Ti64 may be subjected to cyclic loading in their service. Therefore, fatigue strength is an important design criterion. Considerable efforts have been made to understand the influence of the as-built surface conditions on the fatigue performance of AM Ti64, particularly in the high cycle fatigue regime [5,6,7,8,9,10]. In powder bed AM Ti64, surface roughness can reduce the fatigue strength by up to three times compared to the same material with machined and polished surfaces [8]. At the same applied stress, surface roughness can lead to an approximately 75% reduction in fatigue life [8]. Initial studies have correlated the surface roughness parameters with fatigue life and found that the average roughness (*R*_a_) and the maximum surface profile height (*R*_t_) could be correlated to fatigue life, where increased *R*_a_ and *R*_t_ values caused fatigue life reduction [9,10,11]. Similar observations were made in another study [12] where the influence of as-built surface roughness on the high cycle fatigue (HCF) performance of Ti64 alloys manufactured via either the electron beam powder bed fusion (EB-PBF) or laser powder bed fusion (L-PBF) processes was studied and found a 35% reduction in fatigue strength (at 10^7^ cycles). However, a correlation between surface roughness parameters and fatigue life has not been found in other materials, such as nickel-based alloys [13]. A more holistic approach by Sanaei et al. [14] found a good correlation between the surface profile peak or trough with the fatigue life of various materials. Kahlin et al. [15] studied the fatigue behaviour of as-built Ti64 using L-PBF and EB-PBF processes containing purposely built notches, finding that surface roughness was the single most severe factor causing fatigue life reduction and derived a fatigue notch factor for the rough surfaces.

Efforts have also been devoted to fatigue life prediction methods, which are the notch stress method [6,16,17] and the fracture mechanics approach [18,19,20,21]. Dinh et al. [17] applied the notch stress method to study the synergistic effects of gas pores and surface roughness on the fatigue life of laser powder bed fusion Ti64. Vayssette et al. [22] used a fracture mechanics-based numerical model in which surface roughness profiles were measured using an optical light interferometer, and this result was used to build a realistic finite element model to mimic the surface condition. However, the fatigue strength was not well predicted since the micro-notches associated with the as-built surface were not well described. Peng and Jones et al. [18,19] treated the surface profile as a series of small cracks for predicting the durability of test samples made of 18Ni 250 maraging steel and Ti64 based on the crack growth life.

Limited research is available in the open literature on surface waviness in WAAM materials. Dirisu et al. [23] studied the influence of as-built surfaces on the tensile and fatigue strengths of a WAAM-made structural steel ER70S-6. Thicker layer deposition caused stairsteps between the layers with a typical distance of 0.18 mm between the peak and trough [23], which caused an approximately 75% reduction in fatigue strength for a given life compared to the samples with machined surfaces. In a similar study on ER50-6 steel, the as-built surface waviness was about 0.14 mm from the peak to the trough [24]. To the authors’ knowledge, there are no published data on the influence of the as-built surface on the fatigue performance of WAAM Ti64. Although surface roughness can be reduced via post-process machining, with an increasing emphasis on sustainability and reducing the buy-to-fly ratio, it is important to reduce the manufacturing effort and material waste, particularly for metals that are either more expensive to purchase or harder to machine. Furthermore, for parts with complex geometry, fully machining the entire rough surface is not always possible, and the effect of partial machining on durability is unknown. In such scenarios, it is important to understand the acceptable level of surface roughness and to develop a method for fatigue life evaluation.

The work reported in this paper aimed to study the influence of as-built surface conditions on the fatigue life of WAAM Ti64. Fatigue tests were conducted under bending loads. Two different predictive methods were used: one based on the traditional notch stress method treating troughs as micro notches and the other based on the fracture mechanics treating troughs as small cracks. The modified Hartman–Schijve equation and small crack growth rate data were employed in the fracture mechanics method. The predicted fatigue lives from both methods were compared with the experimental test results.

## 2. Materials and Methods

A WAAM Ti64 wall 12 mm thick was built by a single bead deposition process. The feedstock was a grade-5 Ti64 wire 1.2 mm in diameter. The plasma arc was used as a heat source, and Argon gas of 99.99% purity was directed precisely at the melt pool to prevent oxidation. The process parameters are listed in Table 1. The dimension of the wall was 300 × 150 × 10 mm^3^ in length, height, and thickness, as shown in Figure 1a.

The wall was cut off from the substrate plate after deposition. Rectangular fatigue samples were extracted by wire electric discharge machining (W-EDM), as shown in Figure 1b,c. To investigate the influence of the as-built surface condition on fatigue life, two types of samples were prepared, i.e., 22 samples with the as-deposited surface (no machining) and 14 with machined and polished surface. For the ‘as-deposited’ samples, one side of the sample was machined to facilitate sample mounting on the bending test frame. In order to observe the crack propagation path during the fatigue testing, some of the as-deposited samples were ground and polished on the ND-TD plane (Figure 1b) using SiC paper and 0.06 µm silica suspension, respectively. The samples were then etched using Kroll’s reagent for approx. 45 s. For samples with the machined surfaces, the as-deposited surface was removed using high-precision milling and was subsequently ground and polished to achieve an average surface roughness of 0.2 µm, as recommended by ASTM E466 [25]. In the following text, samples with the as-deposited surface are called as-deposited samples. Samples with machined and polished surfaces are called machined samples. The mechanical properties of WAAM Ti64 are listed in Table 2, which is based on the published literature of the same material.

**Figure 1 materials-16-05355-f001:**
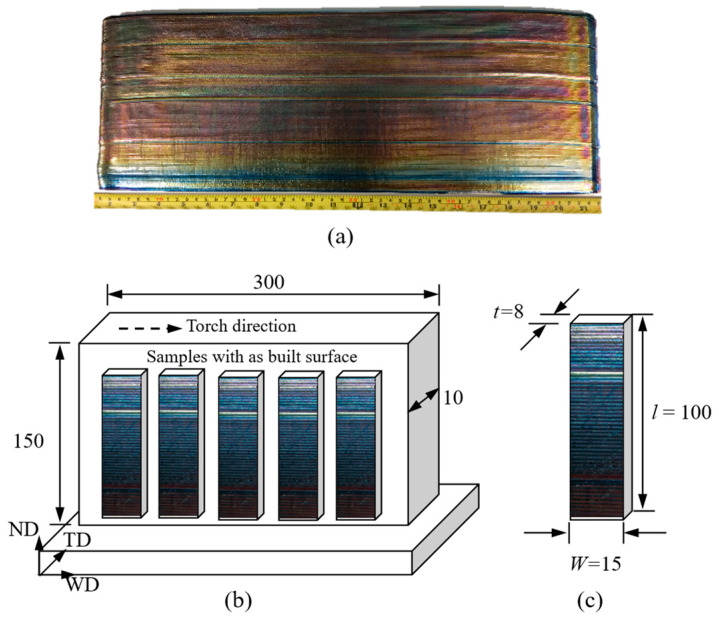
(**a**) Photo of WAAM Ti64 wall built by single-pass deposition method, (**b**) Schematic of sample extraction plan, (**c**) Geometry and dimensions of test sample with as-built surface (unit: mm); WD = weld torch travel direction, TD = transverse direction, ND = normal direction [26].

**Table 2 materials-16-05355-t002:** Material properties of WAAM Ti64.

Modulus (GPa) [4]	Yield Strength (MPa) [4]	Ultimate Tensile Strength (MPa) [4]	Elongation at Failure [4]	$\mathrm{Threshold}\;\mathrm{SIF}\;\mathrm{Range}\;\Delta K_{\mathrm{th}}(\mathrm{MPa}\sqrt{m}$
116	872 ± 16	952 ± 9	17.5 ± 4	4.5

### 2.1. Characterisation of As-Built Surface

The as-deposited surface was characterised using a metrology tool called Formtracer. This is a type of contact mode surface measurement technique where the probe touches and scans the surface (Figure 2a,b). The Mitutoyo FT SV-C3200/4500 series formtracer, with an arm containing a 4 µm diameter diamond tip stylus, was used. During the measurements, a 5 mN load was applied to keep the stylus in contact with the sample surface, as illustrated schematically in Figure 2b. The scanned data were recorded and analysed using the Formtracepak software (by Mitutoyo UK Ltd., Andover, UK), version 5.602. Here, “surface waviness” is quantified by the term “*surface roughness*” using the following representative parameters employing commonly used equations, i.e., *R*_a_ being the arithmetical mean height, *R*_y_ the maximum depth of troughs, and *R*_z_ the average of 10-point surface roughness where ${\left|y_{i}\right|}_{\max}$ and ${\left|y_{i}\right|}_{m\mathrm{in}}$ are the five higher local maxima and lower local minima, respectively, see Figure 2c. These surface roughness parameters are defined by Equations (1)–(3) [20,27].
(1)$$R_{a}=\frac{1}{n}\sum_{i=1}^{n}\left|y_{i}\right|$$
(2)$$R_{y}=\left|y_{\max}-y_{\min}\right|$$
(3)$$R_{z}=\frac{1}{5}\left[\sum_{i=1}^{5}{\left|y_{i}\right|}_{\max}+\sum_{j=1}^{5}{\left|y_{i}\right|}_{\min}\right]$$

However, these conventional surface roughness parameters are insufficient for fatigue life prediction due to the following reasons. For the notch stress method, the notch base radius and notch mouth profile are also required to calculate the stress concentration factor (*K*_t_). For the fracture mechanics-based approach, *K*_t_ is also required if the notch zone plasticity effect should be considered in the stress intensity factor solution. Therefore, the commonly used surface roughness parameters (*R*_a_, *R*_y,_ and *R*_z_) are only used to characterise the as-deposited surfaces. For fatigue analysis, a “notch” profile (Figure 3a) representing the typical surface waviness was further characterised in terms of the notch depth (*d*), the notch mouth opening angle (*θ*), and notch base radius (*r*), as shown in Figure 3b. The parameter *r* was determined using a polynomial equation derived from the spline constructed by the data points obtained by the Formtracer. The spline was then used to compute the first and second derivatives, *f*′(*z*), *f*″(*z*), which are used in Equation (4) [28].
(4)r=(1+f′z2)32f″(z)

### 2.2. Fatigue Testing

A three-point bending (3-point-bending) fatigue test was conducted on a 10 kN Instron servo-hydraulic test machine. The test was performed under a constant amplitude load-controlled condition. A standard sinusoidal waveform was applied with a cyclic load ratio *R* = 0.1 and a loading frequency of 10 Hz. The experimental setup is shown in Figure 4. The maximum tensile stress *S*_max_ acting on the specimen’s lower surface was calculated by Equation (5) [29].
(5)$$S_{\max}=\frac{3PL}{2Wt^{2}}$$
where *P* is the applied load, *L* the distance between the supporting rollers (*L* = 60 mm), and *W* and *t* are the width and thickness of the sample, respectively, see Figure 1c.

### 2.3. Life Prediction Methods

Two different methods have been used for predicting the fatigue life of bending fatigue test samples. One is based on the traditional method for durability analysis using the material S-N data in conjunction with the notch stress concentration factor (*K*_t_) arising from the surface roughness; the other is based on the fracture mechanics approach, treating the notch as a crack and using the stress intensity factor range (Δ*K*) and the material’s fatigue crack propagation rate property. The analyses of *K*_t_ and Δ*K* are presented in Section 3.

For the fracture mechanics method, small crack growth rate data were used based on our previous work [30], in which the crack growth rates of short and long cracks were obtained for the same material, which is shown in Figure 5. The results indicate that small cracks exhibited a higher crack growth rate than long cracks under the same applied Δ*K*. The observed difference in crack growth rates could be attributed to the difference in constraints imposed by the elastic material surrounding the crack, which can vary in the case of small cracks [31]. The constraints experienced by small cracks that initiate from or grow on a free surface of smooth samples differ from those experienced by through-thickness long cracks in the same material [32]. Consequently, even when the loading conditions and crack sizes follow the Linear Elastic Fracture Mechanics (LEFM) requirements, the physically small cracks exhibit faster crack growth rates. Furthermore, long cracks usually have longer plastic wake that reduces the crack growth rate due to the crack closure effect [31].

The modified Hartman–Schijve equation [33,34,35,36], Equation (6), was used to include the short crack growth as shown in Figure 5.
(6)$$\frac{da}{dN}=D{\left(\frac{\Delta K-\Delta K_{\mathrm{thr}}}{\sqrt{1-K_{\max}/A}}\right)}^{p}$$
where *K*_max_ is the maximum stress intensity factor, Δ*K* is the stress intensity factor range, *A* the cyclic fracture toughness, and *D* and *p* are material constants. The term Δ*K*_thr_ is the effective threshold of the stress intensity factor range for small cracks. For long cracks, threshold Δ*K*_th_ is the value of applied Δ*K* corresponding to a crack growth rate of 10^−10^ m/cycle according to the ASTM E647 standard [37].

Work in [19,38,39,40,41] has demonstrated that the variability in crack growth rates can be modelled using the modified Hartman–Schijve equation. In this work, Equation (6) was used to compute the variability in the crack growth rates in WAAM Ti64, where the material constants *D* and *p* were obtained from long crack test data [30]. The values suggested by [34,42] may be suitable for microstructurally small cracks but may be impractical for physically small cracks when initiated from defects found in WAAM Ti64, as the work in [43,44] found higher Δ*K*_thr_ values for small cracks in Ti64 at 1.83.0 MPa$\sqrt{m}$. In this study, Δ*K*_thr_ = 2 MPa$\sqrt{m}$ was used in Equation (6) to predict the small crack growth behaviour in the material; the prediction curve is shown in Figure 5.

## 3. Finite Element Analysis of Stress Concentration Factor and Stress Intensity Factor

### 3.1. Stress Concentration Factor (K_t_)

The *R*_a_, *R*_y_, and *R*_z_ values measured from the as-deposited surface were 26.8 ± 2.8 µm, 245 ± 29 µm, and 152 ± 15 µm, respectively, representing the overall characteristic of the specimen surface.

In the case of WAAM material used in this study, a *unit* of the periodically repeat surface undulation is called a *notch*, which is characterised by three parameters, i.e., notch base radius (*r*), notch depth (*d*) and notch mouth opening angle (*θ*); all were obtained from the Formtracer measurement data (Figure 3a). Corresponding to their definitions in Figure 3b, parameters *r*, *d*, and *θ* can fully characterise the profile of a trough or a notch. In the stress analysis and fatigue life prediction, the notch profile at the beam mid-span was used. Typical value ranges included *r* = 90–200 µm, *d* = 50–320 µm, and *θ* = 167°–175°.

In the literature, it has been observed that with the ratio *r*/*t* < 0.03 and *t*/(*t−d*) < 1.05, the notch mouth angle *θ* becomes the dominant factor on the *K*_t_; hence, it is termed as *K*_tθ_ from here onwards [29]. A similar trend was also found in this study where *K*_tθ_ decreased as *θ* increased (with *r*/*t* and *t*/(*t*−*d*) kept constant) until *θ* reached 180° (*K*_t_ = 1).

In the FE analysis, the stress concentration factor *K*_tθ_ was calculated by the ratio of maximum local stress at the notch root to the applied bending stress at the lower surface of the beam mid-span. The as-deposited surface profile from metrology data was imported into the ABAQUS software Version 6.14 using the data points from Formtracer, and the value of *K*_tθ_ was calculated, as shown in Figure 6. It was observed that due to the asymmetric nature of the notch, the difference between a symmetric V-notch and the actual notch was within ~7%. Nevertheless, the *K*_tθ_ value is still dependent on the *θ* angle. To reduce the computational time, the 60,000 data points obtained from Formtracer in a single scan were reduced to 6000 by removing intermediate points without affecting the *K*_tθ_ values. The data points were converted into splines which were then converted to 2D geometry models using the software package CATIA V5. The profile was then used as a surface model to generate the finite element geometry in ABAQUS. A linear elastic material model was considered with the plane strain condition. An element size of 0.02 mm was used near the notch root, which gradually increased to 0.5 mm away from the notch. The element size was selected after mesh sensitivity analysis for solution convergence. The load and boundary conditions applied are shown in Figure 6.

To verify the finite element analysis, Equation (7) was used [29]:(7)$$K_{t\theta }=1.11K_{t}-\left[-0.0159+0.2243\left(\frac{\theta }{150}\right)-0.4293{\left(\frac{\theta }{150}\right)}^{2}+0.3609{\left(\frac{\theta }{150}\right)}^{3}\right]{K_{t}}^{2}$$
where *K*_tθ_ is the notch stress concentration factor taking into account the notch mouth angle *θ*, and *K*_t_ is the stress concentration factor of a straight-sided U notch with a semi-circular base.

For a typical notch in the as-built surface, the *K*_tθ_ value was between 1.25 and 1.85. The difference between the FEA and the analytical solution from Equation (7) was below 5%.

### 3.2. Stress Intensity Factor in Mode-I Loading (K_I_)

For the analysis based on fracture mechanics, the mid-span notch is treated as a crack, as shown in Figure 7a. To verify the FE model, the *K*_I_ analytical solution for a standard single-edge notch specimen under bending, i.e., the SEN(B) configuration, was used [45,46]. The ABAQUS code was used for FE analysis. The SEN(B) sample was modelled using the two-dimensional (2D) model. The width of the sample was 8 mm. The fracture surface analysis of the as-built sample in Figure 8 showed no significant variation in the crack front across the width of the sample. Therefore, it is reasonable to assume a uniform crack front and analyse the crack propagation problem with 2D plane strain elements (CPE4R). The benchmark sample was considered symmetrical about the crack depth; therefore, the symmetrical boundary condition was applied with half the beam being modelled. The applied load on the half model was *P*/2 and was applied on the top surface as a point load. Using Equation (5), the tensile stress acting on the lower surface of the beam was calculated as 380 MPa. Since the notch at mid-span had the highest stress concentration, and almost all the samples failed at the middle of the sample, only the centre notch was modelled. The actual profile was then modelled with an initial notch depth of 0.05 mm and a crack length of 0.1 mm. The element size at the crack front was 0.006 mm, which was progressively increased to 0.5 mm for both models and was selected after the mesh convergence study (with an accepted margin of error of <3%). Linear elastic material properties were also used for this analysis, and the displacement extrapolation method was adopted to calculate *K* ahead of the crack tip.

The modelling result was verified with an analytical solution for three different crack lengths in Figure 7b. The difference between the analytical solutions and FE analysis was below 3%. Therefore, Equation (8) from [45] was used in this study.
(8)K=PLWt322.9at12−4.6at32+21.8at52−37.6at72+38.7at92
where *P* is the applied load, *L* the distance between the supporting rollers, *W* the specimen width, *t* is the thickness, and *a* is the crack length.

## 4. Experimental Test Results

The stress vs. life (S-N) data of as-deposited and machined surface samples under a bending fatigue load is presented in Figure 9. It shows a large reduction in fatigue strength at a given life, the worst case being a 50% decrease in strength at 3 × 10^5^ load cycles. The fatigue life was significantly reduced under the same applied stress, e.g., life was reduced by 10 times at an applied stress of 600 MPa owing to the stress concentration arising from the surface waviness, resulting in a premature crack initiation at the notch roots. In this study, fatigue cracks always initiated from a single ‘notch-like’ feature that experienced maximum tensile stress during the bending test.

As mentioned in [29], three parameters, *r*, *d* and *θ* determined the notch profile and the stress concentration factor value for the specimen. Therefore, after sample failure, the crack initiating the notch was identified and traced using a recorded surface measurement. This enabled the determination of *r*, *d*, and *θ*, values of the crack initiating notch in all the as-deposited samples.

Figure 9 also shows that the scatter in test data varied at different applied stress levels. In the machined samples, scatter was the smallest at an 800 MPa applied bending stress, as the stress level was close to the yield strength of the material. The scatter increased as the applied stress decreased. This is because, at higher stress levels, the surface condition was less likely to influence crack nucleation, as micro-cracks were formed much earlier in the fatigue life, followed by the crack growth, as shown for axially loaded samples in [47]. This could be the reason for the lower scatter in all the samples tested at 800 MPa. At lower applied stress levels, the surface condition became more relevant, and crack initiation depended on the surface irregularities, which varied among the test samples; hence, a larger scatter was observed for the machined samples tested at 600 MPa. On the other hand, as-deposited samples were tested at lower applied stress, i.e., at 600 MPa, it showed less scatter than the machined samples. As the stress concentration factor was similar in all the as-deposited samples, the fatigue life scatter was mainly dominated by crack growth from the crack initiating the centre notch, whereas fatigue life was dominated by crack initiation life in the machined sample. Hence, a lower scatter was found in the as-deposited samples.

## 5. Fatigue Life Prediction

### 5.1. Notch Fatigue Approach

The first prediction method was based on the traditional notch strength approach [48] with the following assumptions: (a) the surface undulation can be considered as a series of individual notches that act as stress raisers; (b) the predicted fatigue life represents the life to crack initiation life from a trough, and the subsequent crack propagation life is neglected for the small laboratory samples.

In the notch stress method, fatigue strength is estimated by the material S-N data and the notch root stress concentration factor (*K*_t_). In the first cycle, fatigue strength is considered equal to the material’s yield strength [49]. At one million cycles, fatigue strength can be determined by reducing the smooth component’s fatigue strength by a factor of *K*_t_ [49]. Connecting these two points at the first cycle and a million cycles, an S–N curve was established for the notched specimen with the *K*_t_ effect. This is an empirical method that can provide an estimation of the notch fatigue strength.

In the prediction, the applied maximum bending stress, *S*_max_, at 3 × 10^5^ cycles was reduced by a factor of *K*_tθ_, which had a minimum value of 1.25 and maximum value of 1.85, according to the FE analysis in Section 4. Hence, the two predictive curves in Figure 10 represent a range of the fatigue life for the as-built samples.

Using the most severe notch, *K*_tθ_ = 1.85, life prediction agreed with test data only at the highest applied bending stress (600 MPa). Life was overestimated by up to one magnitude for lower stresses. The slopes of the prediction curves were shallower than the test data. The poor agreement between the test and prediction could be explained as follows. The *K*_t_-based method works better if the crack initiation life is dominant, i.e., a large portion of the total life is spent in the crack initiation stage from a notch root. In the bending test, the notch tip is under a much higher mode-I stress than compared to axial loading; hence, the crack initiation life is much shorter or negligible, and the crack propagation stage dominates. In this study, we investigated the behaviour of the crack growth that is shown in Figure 11, which illustrates a portion of the specimen height and an earlier part of the crack length, which were measured during the tests. Based on the definition that the total life (*N*_t_) is the sum of the crack initiation life (*N*_i_) and the subsequent crack propagation life (*N*_p_), we found that the crack initiation life occupied approximately 2–8% of the total life. The cycle numbers displayed in Figure 11 represent the accumulated cycles from the first cycle at the corresponding crack length measurements.

### 5.2. Fracture Mechanics Approach

The second prediction method was based on the fracture mechanics approach, treating a notch (a trough) as a small crack that propagated under cyclic loads. The fatigue crack growth rate was calculated using Equation (6) and using the material constants obtained in [30] as *D* = 1 × 10^−10^, *p* = 2.5, Δ*K*_thr_ = 2, and *A* = 90 (in unit of MPa, m). The fatigue life for a given crack length was then calculated by integrating the crack growth rates. The prediction starts from an equivalent initial flaw size (EIFS), which is the notch depth *d* and finishes at a critical crack length and is based on the fracture toughness of the material. The final critical crack length was also established by examining the fracture surfaces, as shown in Figure 8b, which was estimated to be around 4 mm: about half of the sample thickness. Figure 12 shows the flowchart for fatigue life prediction based on fracture mechanics.

Crack growth from a trough was also monitored during the fatigue test (Figure 11), which was then plotted as the crack length (*a*) vs. the load cycle number (*N*) relation in Figure 13a. Under the same applied load, Test-1 lasted longer than Test-2 owing to different values of the maximum applied bending stress; however, the crack growth rate was similar, as indicated by similar slopes of the two *a* vs. *N* curves.

This prediction is also presented in terms of an S-N graph in Figure 13b. Using two different initial crack lengths, representing the depths of the shallowest notch (50 µm) and the deepest notch (320 µm), the predicted life formed an upper bound and a lower bound of the fatigue test data; hence, it can be regarded as a good prediction.

Because the troughs on the as-built surface look like small notches, two different approaches were chosen for life prediction, i.e., the notch stress and the fracture mechanics methods. Predictions in Figure 10 use the notch stress method, showing poor agreement with the test result, which could be attributed to early crack initiation from the troughs; hence, with the crack propagation phase being dominant, i.e., the main mechanism was not crack initiation. Consequently, the slope of the curves did not match the trend slope of the experimental test data. Nevertheless, this is a meaningful exercise and worth reporting in this paper, as the notch stress approach is a familiar engineering method for fatigue prediction at stress concentration sites. Whilst the method works well for large notches, e.g., holes, it does not work for sharp troughs as the crack initiation life was very brief; therefore, the damage process is fatigue crack propagation. On the other hand, the fracture mechanics approach gives a much better prediction (Figure 13). The comparison and contrast of the two prediction approaches (Figure 10 and Figure 13) show the capabilities and limitations of the current methods, as both are used by the research community and industry. In fact, many people may choose the notch stress approach as the troughs look like notches. Here, we wanted to demonstrate that the fracture mechanics approach works better for fatigue cracks initiated from the troughs.

## 6. Conclusions

This research was aimed at quantifying the effect of the as-built surface condition on fatigue life in a wire arc additive manufactured titanium alloy Ti-6Al-4V (WAAM Ti64). Two different methods were used for fatigue life prediction based on the notch-stress or fracture mechanics approaches. Three-point bending fatigue tests were conducted to validate these predictions. Based on this work, the following conclusions have been drawn:Troughs on the as-built surface of WAAM materials can be treated as a series of individual micro notches and can be characterised by three parameters: notch depth, notch base radius, and notch mouth opening angle.The traditional notch stress method cannot predict the correct S–N curve trend slop, which is attributed to the early crack initiation from the troughs and crack propagation as the dominant failure mechanism.The fracture mechanics approach has given good predictions by treating the troughs as small cracks. The predictive method can help with decision making as to whether surface machining is necessary, according to the service load level and fatigue strength target.The future perspective of this work is to use the short crack growth model as an engineering tool for the evaluation of the fatigue strength and for components with surface roughness or crack-like defects.

## Figures and Tables

**Figure 2 materials-16-05355-f002:**
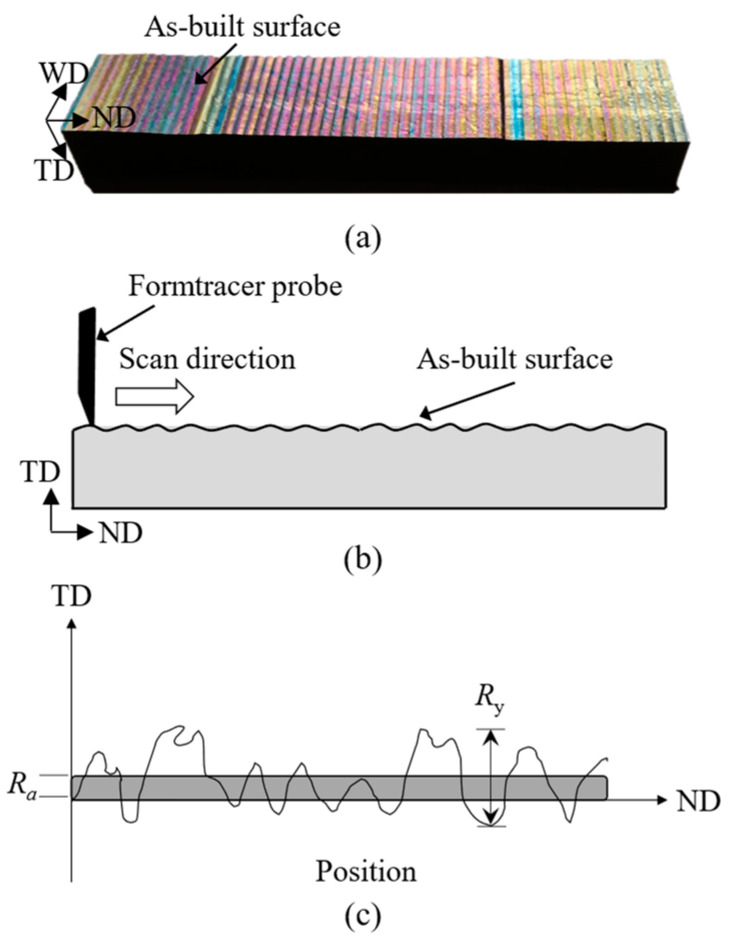
(**a**) Photograph of a fatigue test sample with as-built surface, schematics of a Formtracer probe scanning sample surface (**b**) and surface roughness parameter definition (**c**) [26].

**Figure 3 materials-16-05355-f003:**
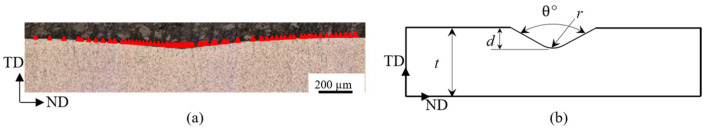
(**a**) High-resolution Formtracer data overlaid onto an optical microscopic image of single waviness (referred as a “notch”) for the depiction of the notch profile, (**b**) Schematic of notch geometry; notch depth (*d*), notch mouth opening angle (*θ*) and base radius (*r*) [26].

**Figure 4 materials-16-05355-f004:**
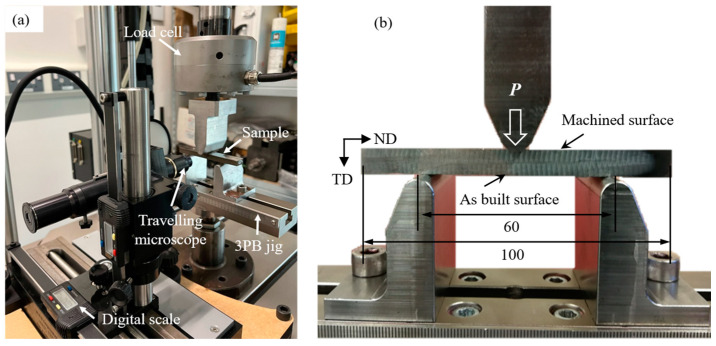
(**a**) Experimental setup showing the test machine, the specimen in 3-point-bending test frame, and the crack monitoring system, (**b**) Close view of the 3-point bending test rig showing the loading and supporting roller positions with the as-built surface facing downwards (unit: mm) [26].

**Figure 5 materials-16-05355-f005:**
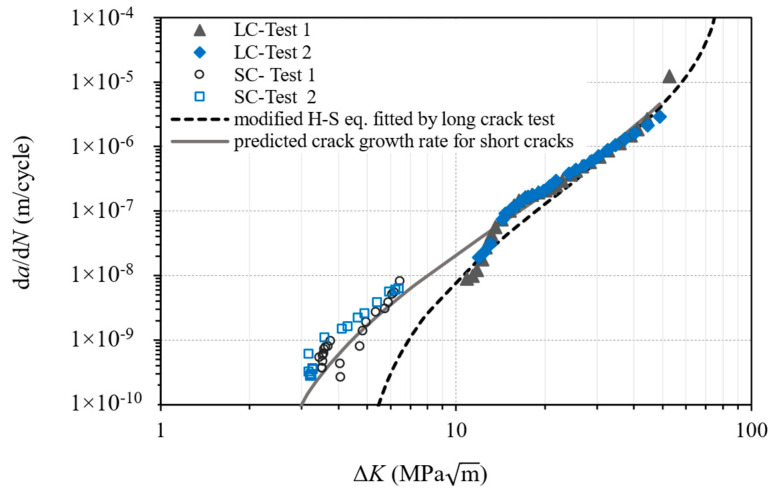
Crack growth rate curves using the modified Hartman–Schijve equation fitted by long crack test data, and a predicted curve for short crack growth using long crack data Δ*K*_thr_ = 2 MPa$\sqrt{m}$, *A* = 90 MPa$\sqrt{m}$ (LC = long crack, SC = small crack).

**Figure 6 materials-16-05355-f006:**
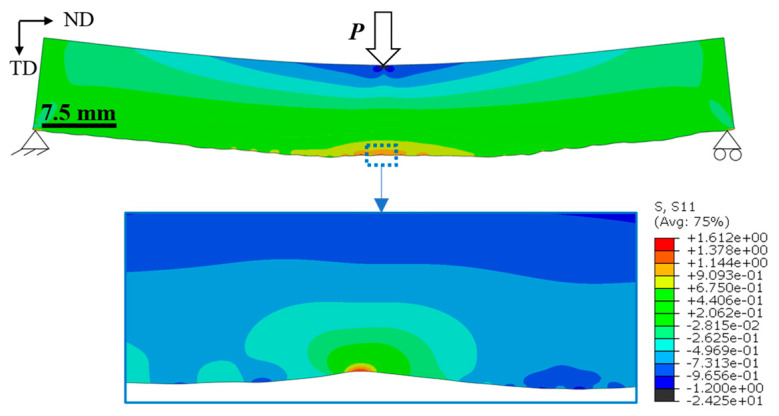
A finite element model for the 3-point-bending specimen, and a zoomed view of the notch at mid-span [26].

**Figure 7 materials-16-05355-f007:**
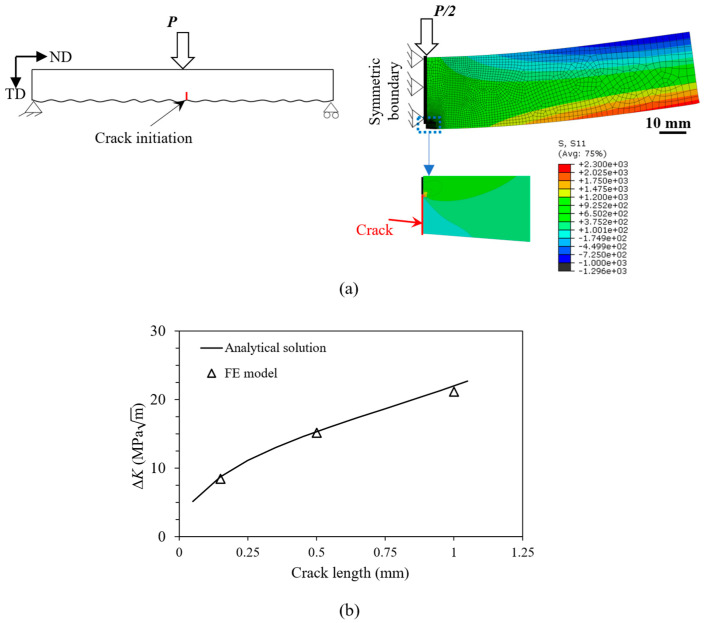
(**a**) Schematic representation and finite element model for calculating the stress intensity factor of a notch at mid-span representing a typical trough on the as-built surface, (**b**) Comparison of stress intensity factor values between finite element model and analytical solution [45].

**Figure 8 materials-16-05355-f008:**
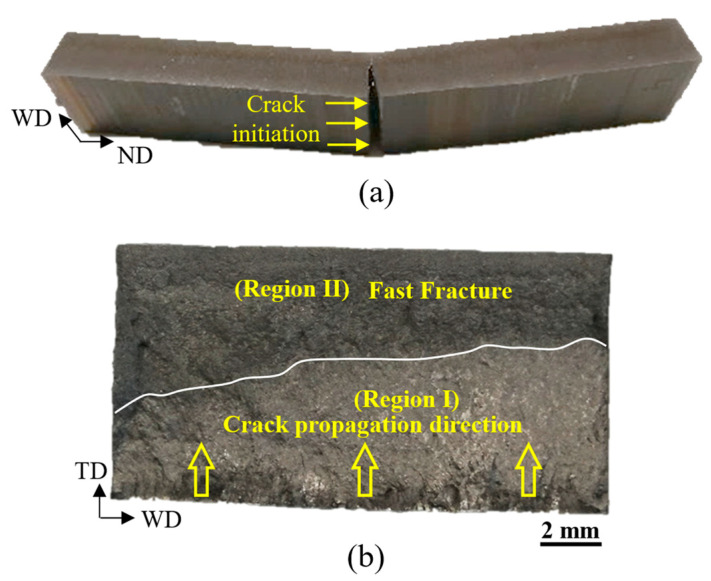
(**a**) Fracture failure at the mid-span of a specimen with an as-built surface, (**b**) Fracture surface showing stable crack growth stage (Region I) and fast fracture (Region II) [26].

**Figure 9 materials-16-05355-f009:**
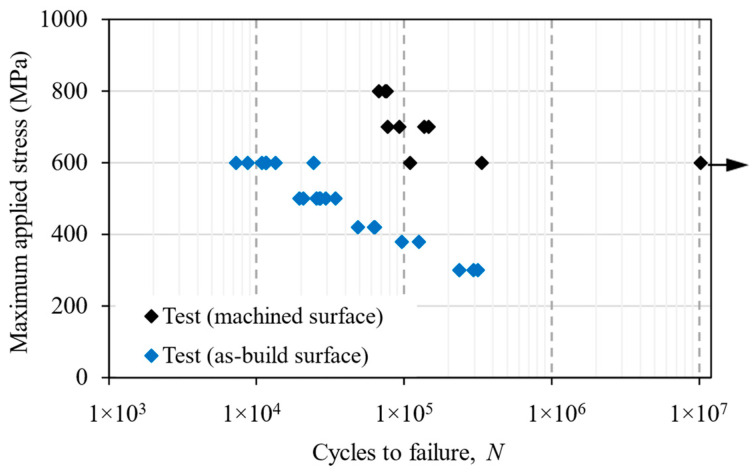
S-N data of 3-point-bending fatigue tests for machined and as-deposited WAAM Ti64 samples. The *y*-axis represents the maximum bending stress on the beam lower surface at the mid-span. The arrow on the far right indicates “test runout”, i.e., the specimen did not fail at this point.

**Figure 10 materials-16-05355-f010:**
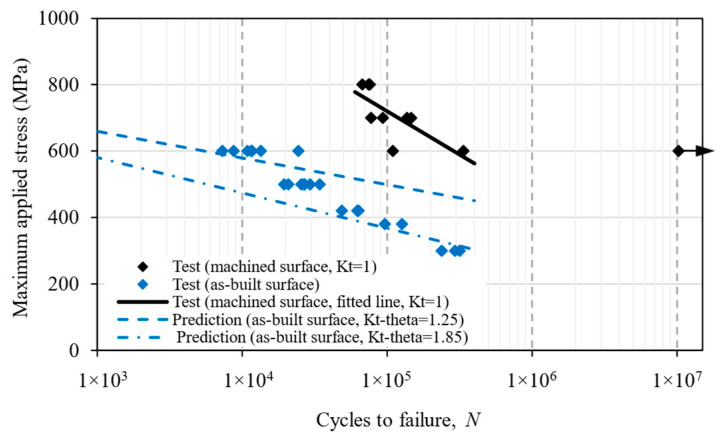
Fatigue life prediction for as-deposited samples using the notch stress approach.

**Figure 11 materials-16-05355-f011:**
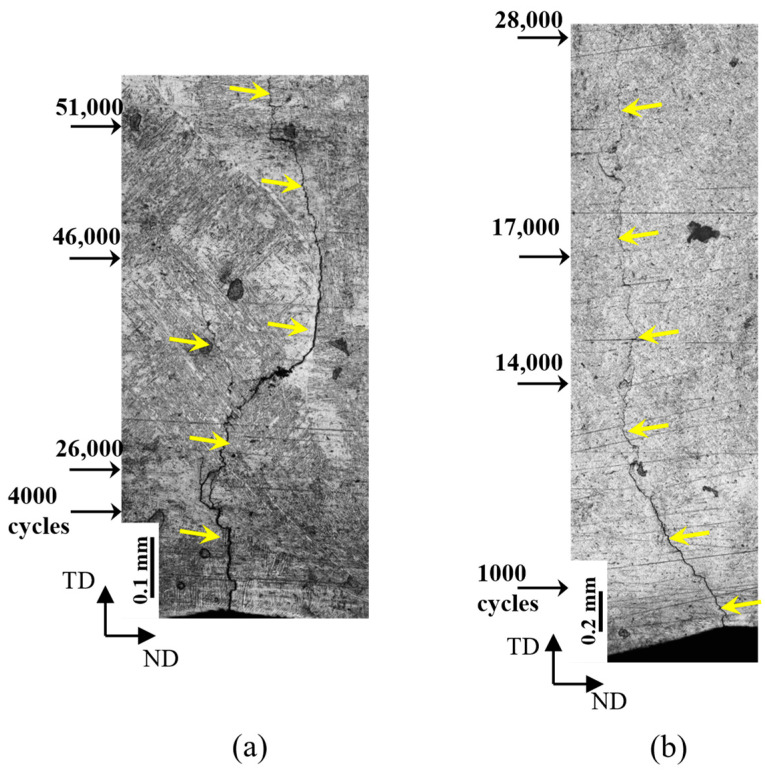
Macroscopic images of the crack propagation trajectories with indicators of accumulated load cycles at crack measurement points: (**a**) 3PB-Test 1 (σ_max_ = 380 MPa, *N*_i_ = 31,000, *N*_p_ = 74,000 and *N*_t_ = 126,737), (**b**) 3PB-Test 2 (σ_max_ = 380 MPa, *N*_i_ = 3000, *N*_p_ = 33,000 and *N*_t_ = 96,875), where *N*_i_ denotes the crack initiation life corresponding to the initiation of a 0.5 mm crack, *N*_p_ is the the crack propagation life from 0.5 mm to around 1.5 mm, and *N*_t_ is the the total life from the first cycle to fracture. The images only captured part of the specimen’s height [26].

**Figure 12 materials-16-05355-f012:**
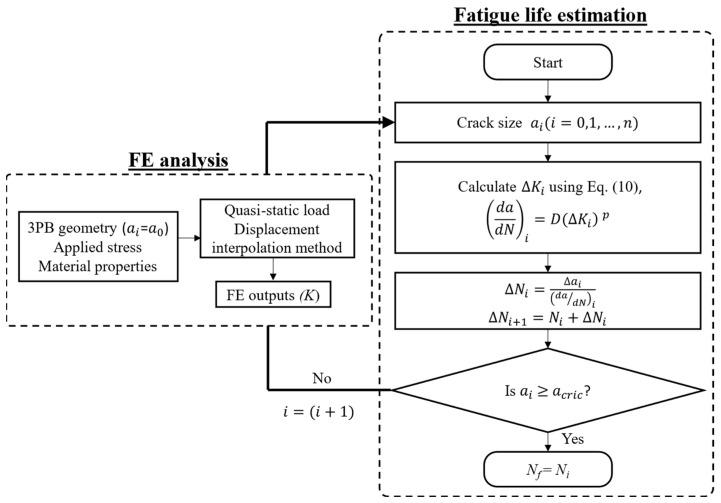
Flowchart for fatigue life prediction used in this study.

**Figure 13 materials-16-05355-f013:**
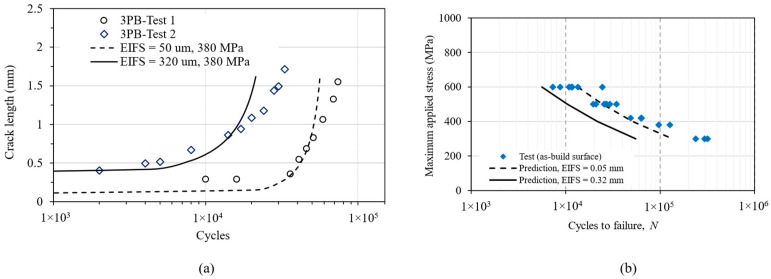
(**a**) Crack length vs. number of cycles for 3-point-bending-Test 1 (σ_max_ = 380 MPa, *N*_f_ = 74,000), 3-point-bending-Test 2 (σ_max_ = 380 MPa, *N*_f_ = 33,000) and predicted fatigue life based on the small crack approach, (**b**) Predicted S–N curves by the modified Hartman–Schijve equation and comparison with experimental data of as-deposited specimens.

**Table 1 materials-16-05355-t001:** Process parameters used for Ti64 wall deposition.

Current (A)	210
Torch stand-off (mm)	8
Torch travel Speed (m/min)	3.5
Wire feed speed (m/min)	2.2
Shielding gas (L/min)	200

## Data Availability

Data will be provided based on the request.

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
