# Peer review of "Predicting the Effect of Surface Waviness on Fatigue Life of a Wire + Arc Additive Manufactured Ti-6Al-4V Alloy"

_materials, 2023, doi:10.3390/ma16155355_

Round 1
Reviewer 1 Report
Dear all, I find the manuscript and the topic interesting. However, some issues have to be clarified or improved.
The following list summarizes my questions and suggestions:
1st Comment: Picture 1(a) does not have enough quality, the illumination is poor, and the photo has a poor resolution. This picture must be improve. In addition, equations 1-3 does not have references, please add the respective references.
2nd Comment: Some very important data are missed in the manuscript. Information about the material property is very important, and it is not mentioned within the manuscript. Data such as Sut (Ultimate tensile strength), Elastic module (very important for FEA) and Sy (Yield strength) are very important to understand the material behavior. On the other hand, for the Fracture mechanics model the DKth for long cracks is also necessary in order to understand the results. Please, provide all these data within the manuscript.
3rd Comment: The “traditional¨ method used in the present research to predict the notch effect seems to be not appropriate. The Fig. 10 shows the model prediction lines overestimating the fatigue resistance of the specimens, which is somewhat weird, because usually the traditional methods for estimating the notch effect underestimate the fatigue resistance as you can see in the reference: Dowling NE. Mechanical behavior of materials: engineering methods for deformation, fracture and fatigue. Prentice Hall; 2007. A suitable approach such as Peterson, Neuber, TDC (Taylor D. The theory of critical distances: a new perspective in fracture mechanics. Elsevier; 2007), Gradient method (Siebel E, Stieler M. Ungleichförmige Spannugsverteilung bei schwingender Beanspruchung. VDI-Zeitschrift 97 1955; p. 121-26) and so on, have to be introduced in the manuscript.
4th Comment: The possibility of local plastic deformation taking place at high stress levels is not considered. The scatter reduction discussion introduced from line 277 have to consider the possibility of having local plastic deformation in the specimens tested at short lives.
5th Comment: The crack front showed in Fig. 8, is much larger on the right side than the left side, and the difference in percentage is more than 60%. This deference is too much for considering a straight front crack. In addition, the cracks that initiate and grow from small irregularities usually initiate its propagation as elliptical or semi-elliptical shapes that finally its front can become straight, therefore the reviewer recommend strongly to introduce a 3D FCG model instead of a 2D model.
6th Comment: My last comment is about the Fracture mechanics model used to predict the fatigue lives of the tested specimens. The stress gradient for the selected three-point bending notch specimen is the result of overlapping the stress gradient due to the bending effect and the stress gradient due to the notch effect, I see that the effect of the first one can be taken by selecting the appropriate DK (for an edge crack growing in a three-point bending cracked specimen), however, the notch effect in the stress field is not taken into account in the model. This effect is very strong when the crack is tiny, i.e., when a << a0, where a0 is the intrinsic crack length that denotes the change between short and long cracks. My question is: How the proposed model take into account the notch gradient effect? Is not necessary to modify the model to introduce this effect?
Please check the introduction and change: as built surface by as-built surface.
Please check the english language in manuscript in general.
Author Response
Reviewer #1: Dear all, I find the manuscript and the topic interesting. However, some issues have to be clarified or improved. The following list summarizes my questions and suggestions:
1st Comment: Picture 1(a) does not have enough quality, the illumination is poor, and the photo has a poor resolution. This picture must be improve. In addition, equations 1-3 does not have references, please add the respective references.
Authors: resolution for Figure 1 is improved and references for equations (1-3) are added.
2nd Comment: Some very important data are missed in the manuscript. Information about the material property is very important, and it is not mentioned within the manuscript. Data such as Sut (Ultimate tensile strength), Elastic module (very important for FEA) and Sy (Yield strength) are very important to understand the material behavior. On the other hand, for the Fracture mechanics model the DKth for long cracks is also necessary in order to understand the results. Please, provide all these data within the manuscript.
Authors: thanks for the good suggestion. Material properties are added to Section 2, as new Table 2.
3rd Comment: The “traditional¨ method used in the present research to predict the notch effect seems to be not appropriate. The Fig. 10 shows the model prediction lines overestimating the fatigue resistance of the specimens, which is somewhat weird, because usually the traditional methods for estimating the notch effect underestimate the fatigue resistance as you can see in the reference: Dowling NE. Mechanical behavior of materials: engineering methods for deformation, fracture and fatigue. Prentice Hall; 2007. A suitable approach such as Peterson, Neuber, TDC (Taylor D. The theory of critical distances: a new perspective in fracture mechanics. Elsevier; 2007), Gradient method (Siebel E, Stieler M. Ungleichförmige Spannugsverteilung bei schwingender Beanspruchung. VDI-Zeitschrift 97 1955; p. 121-26) and so on, have to be introduced in the manuscript.
Authors: thanks for good suggestions. Our response is organised in the following three points.
- the reason for choosing the notch stress method was the troughs of the waved surface look like micro notches rather than cracks, so we compared it with the fracture mechanics approach.
- Figure 10 is corrected (appended below) as the originally submitted figure had error by including the test runout into the regression analysis for producing the best fitted line. In the corrected Fig. 10, the poor agreement between prediction and test is attributed to the early crack initiation from the troughs therefore the subsequent crack propagation phase was dominant, i.e., the mechanism is not crack initiation. Consequently, the trend (slop) of the notch samples S-N was not correctly predicted.
- One of the papers that you suggested is now added as reference [50] by N.E. Dowling. We would say that these papers studied much larger notches with regular radius, such as holes, whereas we idealised troughs as notches which behave between a notch (crack initiation dominant) and a crack (crack propagation dominant).
- Figure 10 and related discussion are revised, and also related text in Abstract (in blue).
Figured 10 is corrected by excluding the “test runout” point in regression analysis.
4th Comment: The possibility of local plastic deformation taking place at high stress levels is not considered. The scatter reduction discussion introduced from line 277 have to consider the possibility of having local plastic deformation in the specimens tested at short lives.
Authors: the notch stress approach is based on Kt, which is a linear-elastic parameter. If the fatigue strength reduction factor, Kf, was used to replace Kt, then plasticity effect is considered to some extent but only empirically. The linear-elastic fracture mechanics (LEFM) approach is also elastic unless the cyclic plasticity induced crack closure is considered. We admit this is a limitation of this paper.
5th Comment: The crack front showed in Fig. 8, is much larger on the right side than the left side, and the difference in percentage is more than 60%. This deference is too much for considering a straight front crack. In addition, the cracks that initiate and grow from small irregularities usually initiate its propagation as elliptical or semi-elliptical shapes that finally its front can become straight, therefore the reviewer recommend strongly to introduce a 3D FCG model instead of a 2D model.
Authors: we agree with you on the limitations of using a 2D model. Since the magnitudes of the peaks and troughs of as-built surfaces vary randomly, we have attempted to use an average trough depth as the initial crack length and 2D model to provide an engineering evaluation of the endurance life. 3D detailed models are better, but it will have to cover a range of variables.
6th Comment: My last comment is about the Fracture mechanics model used to predict the fatigue lives of the tested specimens. The stress gradient for the selected three-point bending notch specimen is the result of overlapping the stress gradient due to the bending effect and the stress gradient due to the notch effect, I see that the effect of the first one can be taken by selecting the appropriate DK (for an edge crack growing in a three-point bending cracked specimen), however, the notch effect in the stress field is not taken into account in the model. This effect is very strong when the crack is tiny, i.e., when a << a0, where a0 is the intrinsic crack length that denotes the change between short and long cracks. My question is: How the proposed model take into account the notch gradient effect? Is not necessary to modify the model to introduce this effect?
Authors: thank you for raising a good point on the influence of the notch stress gradient on the crack tip stress intensity factor (SIF). This effect is more pronounced when crack starts at the notch root and is very small, so the notch stress concentration factor (Kt) and stress distribution (gradient) will affect the SIF. There are two approaches to determine SIF: (a) using the notch local stress (Kt x applied stress) and the actual crack length (excluding the notch size); (b) using the remote applied stress and the crack length includes the notch radius. Our model belongs to the latter, i.e., the trough depth being part of the crack length.
FYI, a figure from Suresh’ book "Fatigue of Materials" ( P. 553) is attached (file name: "Notch stress gradient effect on stress intensity factors"). In the graph, a numerical solution is compared with two analytical solutions, showing that when crack is very small compared to the notch, i.e., the crack is still in the notch plastic zone, using local stress value (Kt x applied stress) is more realistic. However, as the crack grows longer, including the notch in the total crack length gives better approximation.
Comments on the Quality of English Language
Please check the introduction and change: as built surface by as-built surface.
Please check the english language in manuscript in general.
Authors: “as built” is changed to “as-built” (surface), and revised manuscript has been proofread.

Reviewer 2 Report
The paper entitled “Predicting the effect of surface waviness on fatigue life of a wire + arc additive manufactured Ti-6Al-4V alloy” has presented a scientific approach for predicting the fatigue life of WAAM Ti64 alloy based on surface waviness. The manuscript is well-written with a very easy-to-follow flow of information and few typos or grammatical mistakes. However, the outcome of the research is neither reliable nor promising.
I would recommend a major review of the manuscript before being published in “Materials” It is suggested that the authors address the following comments.
1. The claims made throughout the manuscript are required to be cited. For instance: “When considered the mechanical properties, yield and ultimate tensile strengths of WAAM Ti64 are comparable with conventional wrought and considerably higher than the cast materials.”
2. It is recommended to mention the complete word correctly when using an acronym. Line 55: “electron beam (EBLBF) and laser powder fusion (LPBF)”.
3. It is recommended that the authors provide a real photo or an schematic of the experimental setup so that the reader can have an instant understanding of the process.
4. It is advised to avoid long sentences to avoid confusion. For instance, lines 135-137: “However, the conventional surface roughness parameters do not correlate well with the fatigue life 135 of bending specimens as the surface waviness at sample’s mid-span plays a more dominant role 136 than owing to the maximum bending stress and stress gradient through the specimen’s height”. The sentence is also grammatically wrong.
5. The manuscript has made claims without supporting or discussing them. For instance: “the conventional surface roughness parameters do not correlate well with the fatigue life 135 of bending specimens”. The authors need to elaborate this.
6. The procedure and the reason for the chosen approach for determining in Section 3.1 need to be elaborated.
7. It seems that the so-called “fracture surface analysis” in Section 3.2 is only based on simple images which are not reliable. It is recommended that authors consider fractography.
8. The results of fatigue life predations are not in good accordance with the experiments. The approach needs to be revised, considering the contributing factors to crack propagation.
English has an issue. Please do the proof-reading again.
Author Response
Reviewer 2
The paper entitled “Predicting the effect of surface waviness on fatigue life of a wire + arc additive manufactured Ti-6Al-4V alloy” has presented a scientific approach for predicting the fatigue life of WAAM Ti64 alloy based on surface waviness. The manuscript is well-written with a very easy-to-follow flow of information and few typos or grammatical mistakes. However, the outcome of the research is neither reliable nor promising.
I would recommend a major review of the manuscript before being published in “Materials”. It is suggested that the authors address the following comments.
- The claims made throughout the manuscript are required to be cited. For instance: “When considered the mechanical properties, yield and ultimate tensile strengths of WAAM Ti64 are comparable with conventional wrought and considerably higher than the cast materials.”
Authors: new reference is added in Section 1 which shows the comparison of mechanical properties of WAAM Ti64 with conventional Ti64, marked in blue. New reference is [4]
- It is recommended to mention the complete word correctly when using an acronym. Line 55: “electron beam (EBLBF) and laser powder fusion (LPBF)”.
Authors: these have been amended, in Page 2, Line 55
- It is recommended that the authors provide a real photo or an schematic of the experimental setup so that the reader can have an instant understanding of the process.
Authors: Figure 4 is now updated: Fig 4a shows the experimental setup including the test machine and crack monitoring system, and Fig 4b is a closer view of the 3-point-bending test rig and specimen.
- It is advised to avoid long sentences to avoid confusion. For instance, lines 135-137: “However, the conventional surface roughness parameters do not correlate well with the fatigue life of bending specimens as the surface waviness at sample’s mid-span plays a more dominant role than owing to the maximum bending stress and stress gradient through the specimen’s height”. The sentence is also grammatically wrong.
Authors: Thank you. This long sentence is revised and marked in blue on Page 4 Lines 138-139, and also copied below for your quick reference:
“However, these conventional surface roughness parameters are insufficient for fatigue life prediction for following reasons. For the notch stress method, the notch base radius and notch mouth profile are required for calculating the stress concentration factor (Kt). For the fracture mechanics-based approach, Kt is also required if the notch zone plasticity effect should be considered in the stress intensity factor solution.”
- The manuscript has made claims without supporting or discussing them. For instance: “the conventional surface roughness parameters do not correlate well with the fatigue life of bending specimens”. The authors need to elaborate this.
Authors: This comment is related to your previous one, or similar to it; please see our reply to your comment No. 4.
- The procedure and the reason for the chosen approach for determining in Section 3.1 need to be elaborated.
Authors: more explanation is added to Section 3.1 and marked in blue, and also copied below for your convenience.
“In the notch stress method, fatigue strength is estimated by the material’s S-N data and the notch root stress concentration factor (Kt). At the first cycle, fatigue strength is considered to be equal to the component's material’s yield strength [46]. At one million cycles, fatigue strength is determined by reducing the fatigue strength of a smooth component the material by a factor of Kt [46]. Connecting these two points at the first cycle and a million cycles will produce an S-N curve for the notch with notch stress concentration factor of Kt. This is an empirical method that can provide an estimation of the notch fatigue strength.”
- It seems that the so-called “fracture surface analysis” in Section 3.2 is only based on simple images which are not reliable. It is recommended that authors consider fractography.
Authors: thanks for the comment. For the intended purpose, we think the macroscopic image in Fig. 8 is sufficient to show the two distinct regions of crack growth, stable crack growth and fast crack growth regimes; both are clearly identified in the photo owing to the clear contrast. Higher resolution SEM images can show better in very localised zones, e.g., crack initiation sites, but will not add new info to the bigger picture.
- The results of fatigue life predations are not in good accordance with the experiments. The approach needs to be revised, considering the contributing factors to crack propagation.
Authors: We think you refer to Figure 10 on fatigue life prediction based on the notch stress approach. Explanation is below, some of it are added to the revised manuscript (Abstract, and Section 5.2, Page 12-13, in blue). Ref [50] is added as the source of the notch stress approach, as suggested by Reviewer #1.
“Because the troughs on the as-built surface look like small notches, two different approaches were chosen for life prediction, i.e., the notch stress and the fracture mechanics methods. Predictions in Fig. 10 used the notch stress method, showing poor agreement with the test result, which is attributed to early crack initiation from the troughs hence the crack propagation phase being dominant, i.e., the main mechanism is not crack initiation. Consequently, the slops of the curves do not match the trend slop of the experimental test data. Nevertheless, this is a meaningful exercise and worth reporting in this paper as the notch stress approach is a familiar engineering method for fatigue life prediction at stress concentration sites. We want to demonstrate whilst it works for large notches, e.g., holes [50], it does not work for sharp troughs (a feature of surface waviness in AM metals). One the other hand, the fracture mechanics approach gives much better prediction (Fig. 13). The comparison and contrast of the two prediction approaches (Figs. 10 and 13) shows the capabilities and limitations of the current methods, as both are used by the research community and industry. In fact, many people may choose the notch stress approach as the troughs look like notches. Here we want to demonstrate that the fracture mechanics approach works better for fatigue cracks initiated from surface roughness/waviness.”
Comments on the Quality of English Language
English has an issue. Please do the proof-reading again.
Authors: proof reading has been done in the revised version.
Reviewer 3 Report
The entitled manuscript 'Predicting the effect of surface waviness on fatigue life of a wire + arc additive manufactured Ti-6Al-4V alloy' represents a numerical and experimental fatigue studies of additively manufactured titanium alloy. The manuscript needs several improvements to be accepted for publication. See the comments below:
- The objective and the application of this study is not clear. More details about the application of this study is needed.
- The abstract and the conclusion must be rewritten to show the novelty of this work.
- The flowchart in Fig. 12 must be modified to understand the strategy.
The English level MUST be improved.
Author Response
Reviewer 3: The entitled manuscript 'Predicting the effect of surface waviness on fatigue life of a wire + arc additive manufactured Ti-6Al-4V alloy' represents a numerical and experimental fatigue studies of additively manufactured titanium alloy. The manuscript needs several improvements to be accepted for publication. See the comments below:
- The objective and the application of this study is not clear. More details about the application of this study is needed.
Authors: Objectives are stated in Section 1, Line 85-88 and one of the Applications is for parts with complex geometries which are difficult to access for machining and polishing of as-built surfaces. For some applications, surface machining and polishing are not required, so assessment of fatigue durability is important to ensure structural integrity. Following paragraph is added to Section 1, Lines 82-89.
“Although surface waviness can be reduced by machining, with an increasing emphasis on sustainability and reducing the buy-to-fly ratios, it is important to reduce the manufacturing effort and materials waste, particularly for metals which are either expensive to purchase or hard to machine. Furthermore, for parts with complex geometries, fully machining of the entire rough surface is not always possible and the effect of the partial machining on durability is unknown. In such scenarios, it is important to understand the acceptable surface waviness and to propose a methodology for fatigue life prediction in the presence of surface waviness.”
- The abstract and the conclusion must be rewritten to show the novelty of this work.
Authors: Abstract and Conclusions are revised, and sentences related novelty are marked blue and copied below for your easy reference.
In Abstract: “Surface machining and polishing may not always be required, as it depends on the applications and service load levels. This research has demonstrated that the fracture mechanics based approach can be used for predicting the fatigue life for WAAM titanium alloys in as-built conditions; hence can be a tool for decision making on the level of surface machining.”
In Conclusions, following two bullet points relate to novelty of the work:
• The traditional notch stress method cannot predict the correct S-N curve trend slop, which is attributed to the early crack initiation from the troughs and crack propagation being the dominant failure mechanism.
• The fracture mechanics model can help with decision making as to whether surface machining would be necessary according to the service load level and fatigue strength target.”
- The flowchart in Fig. 12 must be modified to understand the strategy.
Authors: The flowchart has been modified for better understanding of the strategy.
Comments on the Quality of English Language
The English level MUST be improved.
Authors: proof reading has been done in the revised version.
Reviewer 4 Report
This manuscript presents a study about the fatigue life prediction for the Ti-6Al-4V alloy obtained by additive manufacturing. The effect of the surface waviness is analyzed in detail. Two different approaches are adopted, comparing the predictions with experimental data. The results show that the notch stress method overestimated the fatigue strength, while the fracture mechanics approach provided good predictions.
The relevance of the topic addressed in the paper is clearly described in the introduction of the paper, including references to previous studies on fatigue life in this alloy obtained by additive manufacturing. Besides, the main objective of the paper is clearly described, including the methods used to achieve it.
The process conditions used in the WAAM process to produce the wall (and consequently the samples) must be provided in the paper. This allows to reproduce the experimental procedure for any researcher.
The geometry and dimensions of the machined samples is accurately obtained. On the other hand, the as-deposited samples present one rough face. The impact of this issue in the obtained results can be commented in the paper, particularly the thickness of 8 mm that is not guaranteed.
The surface waviness of the as-built material was evaluated using surface roughness parameters. Since in the three-point bending the maximum tensile stress occurs at mid-span, the surface waviness is important only at this location. Can you elaborate more about it?
The enumeration of figures in the text is wrong from line 335 of page 11. Please check the citation to the figures in the text.
The relevance of the topic addressed in the paper is clearly described in the introduction of the paper, including references to previous studies on fatigue life in this alloy obtained by additive manufacturing. Besides, the main objective of the paper is clearly described, including the methods used to achieve it.
Author Response
Reviewer 4: This manuscript presents a study about the fatigue life prediction for the Ti-6Al-4V alloy obtained by additive manufacturing. The effect of the surface waviness is analyzed in detail. Two different approaches are adopted, comparing the predictions with experimental data. The results show that the notch stress method overestimated the fatigue strength, while the fracture mechanics approach provided good predictions.
The relevance of the topic addressed in the paper is clearly described in the introduction of the paper, including references to previous studies on fatigue life in this alloy obtained by additive manufacturing. Besides, the main objective of the paper is clearly described, including the methods used to achieve it.
Authors: thank you for your positive comments.
The process conditions used in the WAAM process to produce the wall (and consequently the samples) must be provided in the paper. This allows to reproduce the experimental procedure for any researcher.
Authors: process conditions and sample preparations are added on Page 3, and process parameters are in Table 1 which is copied below for your easy referencing.
The geometry and dimensions of the machined samples is accurately obtained. On the other hand, the as-deposited samples present one rough face. The impact of this issue in the obtained results can be commented in the paper, particularly the thickness of 8 mm that is not guaranteed.
Authors: this is a very good point. With surface waviness, thickness of 8 mm is a nominal value. Consistent thickness value is crucial for uniaxial tensile test specimens, e.g., to measure the material properties in terms of the stress vs. strain relationship. However, for bending test, uniform thickness is not as crucial as the tensile tests, because the maximum stress is at the mid-span of the beam and stress is not unfirm through the thickness due to bending moment.
The surface waviness of the as-built material was evaluated using surface roughness parameters. Since in the three-point bending the maximum tensile stress occurs at mid-span, the surface waviness is important only at this location. Can you elaborate more about it?
Authors: you are right that in bending test the maximum tensile stress occurs at beam’s mid-span.
In the case of three-point bending, the maximum tensile stress is at the mid-span of the beam specimen. Therefore, when considering the effect of surface waviness on the fatigue life of the beam specimen, it is primarily important to examine the waviness at the mid-span region. Other areas of the specimen can be neglected as the stresses are much lower. This was confirmed by all specimens; they all failed at the mid-span. Therefore, only the trough or “notch” at the mid-span was modelled using a representative value of the trough depth. In our fracture mechanics model, the shallowest and the deepest troughs were selected as the “equivalent initial flaw size (EIFS)” for the starter crack length. Therefore, the predicted fatigue life in Fig. 13 covers the upper and lower bounds of the fatigue life of the specimens.
The enumeration of figures in the text is wrong from line 335 of page 11. Please check the citation to the figures in the text.
Authors: thanks for spotting this typo, which is now amended
The relevance of the topic addressed in the paper is clearly described in the introduction of the paper, including references to previous studies on fatigue life in this alloy obtained by additive manufacturing. Besides, the main objective of the paper is clearly described, including the methods used to achieve it.
Authors: thank you for your positive comments.
Round 2
Reviewer 1 Report
Dear Authors,
I appreciate the effort that you have done for improving the manuscript. The last version of the manuscript is much better than the initial version, and even if it does not satisfy all my questions, I consider the manuscript is now publishable.
Author Response
Thank you very much for approving our paper. We have gone through the final version and corrected language errors to the best that we can.
Reviewer 2 Report
All issues are well addressed.
It may need to be enhanced.
Author Response
Thank you for approving our paper. We have improved English language in this revised version to the best that we can.
Reviewer 3 Report
The future perspectives of this work should be added at the end of the conclusion Section.
Some language errors exist in the manuscript.
Author Response
Thank you for approving our paper. English language has been improved in this revised version to the best that we can.
As requested, a statement for the future perspective is added to the end of the Conclusions section, highlighted in yellow. It is also copied below for your quick reference.
"The future perspective of this work is to use the short crack growth model as an engineering tool for the evaluation of the fatigue strength for components having surface roughness or crack-like defects."